# Evaluation of Prebiotic Potential of Crude Polysaccharides Extracted from Wild *Lentinus polychrous* and *Lentinus squarrosulus* and Their Application for a Formulation of a Novel Lyophilized Synbiotic

**DOI:** 10.3390/foods13020287

**Published:** 2024-01-16

**Authors:** Marutpong Panya, Chamraj Kaewraemruaen, Phairo Saenwang, Patcharin Pimboon

**Affiliations:** 1Research Group for Biomedical Research and Innovative Development (RG-BRID), College of Medicine and Public Health, Ubon Ratchathani University, Warinchamrap, Ubon Ratchathani 34190, Thailand; phairo.s@ubu.ac.th; 2Department of Science and Bioinnovation, Faculty of Liberal Arts and Science, Kasetsart University, Kamphaeng Saen Campus, Nakhon Pathom 73140, Thailand; faascrka@ku.ac.th; 3College of Medicine and Public Health, Ubon Ratchathani University, Warinchamrap, Ubon Ratchathani 34190, Thailand; patcharin.pi.61@ubu.ac.th

**Keywords:** polysaccharide, probiotic, prebiotic, synbiotic, *Lentinus polychrous*, *Lentinus squarrosulus*

## Abstract

Edible mushrooms, including wild mushrooms, are currently being investigated as natural sources to evaluate their prebiotic potential. This study aimed to evaluate the prebiotic potential of crude polysaccharides (CPSs) extracted from wild *Lentinus squarrosulus* UBU_LS1 and *Lentinus polychrous* UBU_LP2 and their application as cryoprotectants in the freeze-drying process to formulate a novel synbiotic product. Based on fruiting body morphology and molecular identification, two wild edible mushrooms named UBU_LS1 and UBU_LP2 were identified as *Lentinus squarrosulus* and *Lentinus polychrous,* respectively. *L. squarrosulus* UBU_LS1 and *L. polychrous* UBU_LP2 contained high amounts of CPS after hot water extraction. Monosaccharide component analysis showed that CPS_UBU_LS1 and CPS_UBU_LP2 were typical heteropolysaccharides. CPS_UBU_LS1 and CPS_UBU_LP2 showed hydrolysis tolerance to the simulated human gastric acidic pH solution, indicating that these CPSs are capable of reaching the lower gastrointestinal tract. Antioxidant activity determined using the 1,1-diphenyl-2-picrylhydrazyl assay revealed that the CPS_UBU_LS1 and CPS_UBU_LP2 displayed greater antioxidant activity comparable with that of ascorbic acid. It was found that CPS_UBU_LS1 and CPS_UBU_LP2 have a high potential for stimulating growth in all probiotic strains. Moreover, both CPS compounds could possibly be used as cryoprotectants in freeze drying, since the viability of the selected probiotic *L. fermentum* 47-7 exhibited cell survival of greater than 70% after 90 days of storage at 4 °C. These results highlight that wild edible mushrooms *L. squarrosulus* UBU_LS1 and *L. polychrous* UBU_LP2 are potential natural sources of prebiotics and can be applied as cryoprotectants in the freeze-drying process. The crude polysaccharide derived from this study could also be considered as a potent antioxidative compound. Therefore, our study provides evidence to support the application of CPSs from wild edible mushrooms in synbiotic product development and in various functional foods. Finally, further evaluation of these prebiotics, including the determination of the potential rehabilitation of beneficial gut microbes in diseased individuals, is currently being conducted by our research group.

## 1. Background

According to the definition provided by The International Scientific Association for Probiotics and Prebiotics (ISAPP), a prebiotic is defined as “a substrate that is selectively utilized by host microorganisms conferring a health benefit” [1]. Three criteria are used for selecting compounds to be matched with prebiotics: first, resistance to the acidic pH in the human stomach and hydrolysis by enzymatic systems in the lower intestine; second, utilization by intestinal microbiota; and third, selection to stimulate the growth of beneficial microbes, especially probiotic-associated species/strains [2]. Fructo-oligosaccharides (FOSs) and galacto-oligosaccharides (GOSs), well-known prebiotics, have been evaluated for their prebiotic potential in some diseases, including type-2 diabetes [3]. Fermentation of prebiotics by the intestinal microbiota results in the production of various short-chain fatty acids (SCFAs), which are beneficial not only in the local intestine but also in distal organs, such as the brain [4]. The positive immunomodulation of prebiotics is another benefit to the host’s health. For example, the purified fraction of *Inonotus obliquus* polysaccharide induced anticancer activities in mice. It accomplished this by inducing proinflammatory cytokines, promoting the proliferation of splenocytes and lymphocytes, increasing Bax expression, and inhibiting Bcl-2 expression [5]. These immunological cells and molecules facilitate apoptosis in tumor cells. As detailed in a previous report, the hot-water-extracted fractions of *L. polychrous* and *Ganoderma lucidum* polysaccharide exhibited anticancer, antiviral, and anti-inflammatory effects [6]. For these reasons, the isolation and characterization of prebiotics from other natural food sources, including edible mushrooms, is an important research topic.

Historically, edible mushrooms have been consumed by humans as natural foods, food supplements, and medicines. Edible mushrooms consumed as medicines contain not only good nutritional components but also excellent natural bioactive compounds with beneficial effects in humans [7]. These benefits include anti-inflammatory [8], antioxidant [9], antimicrobial [10,11], and immunomodulatory [12] effects, as well as effectiveness in the treatment of particular diseases, including diabetes [13], Alzheimer’s disease [14], coronary heart disease [15], and cancer [16]. Different bioactive compounds, including terpenoids, phenolic compounds, proteins, lipids, and polysaccharides, have been isolated from edible wild mushrooms and cultured mycelia [17]. Among these compounds, polysaccharides have been found to possess a vast array of activities, and there has been recent interest in employing mushroom polysaccharides for the modulation of human intestinal microbiota in particular intestinal-associated disorders or diseases; however, the evidence of these benefits has mostly been verified in some edible mushrooms, including *Agrocybe cylindracea* [18], *Volvariella volvacea* [19], *Agaricus bisporus* [20], *Trametes versicolor* [21], and *Lentinus squarrosulus* [22]. 

In addition to prebiotic properties, synbiotics, mixtures comprising live microorganisms and one or more substrates selectively utilized by host microorganisms that confer health benefits to the host, have been an attractive development [23]. The probiotics *Lactobacillus* and *Bifidobacterium* spp. that are incorporated in a synbiotic formula utilize prebiotics for their proliferation and may produce metabolic by-products, such as antimicrobial compounds and other organics. Therefore, the prebiotic properties of specific probiotic strains should be evaluated. 

Several methods are used to prepare synbiotic products. Freeze drying or lyophilization is one of the most common methods used to prepare synbiotics [24]. However, this method uses deep, low temperature under high pressure, leading to the death of some probiotic populations. To overcome this limitation, it is necessary to use effective cryoprotectants during freeze drying. However, the use of crude polysaccharides extracted from edible mushrooms as cryoprotectants has not yet been reported.

In the present study, we evaluated the prebiotic properties of crude polysaccharides extracted from the wild edible mushrooms *L. squarrosulus* UBU_LS1 and *L. polychrous* UBU_LP2. Three probiotic strains, *Limosilactobacillus fermentum* 47-7, *Lacticaseibacillus rhamnosus* NH9, and *Lacticaseibacillus rhamnosus* 23-2, were used to evaluate growth promotion by crudely extracted prebiotics. Among these strains, *L. fermentum* 47-7 was selected as the model probiotic for synbiotic incorporation. Our results suggest the possibility of using crude polysaccharides from *Lentinus* species not only as prebiotics, but also as cryoprotectants. The prebiotics and formulated synbiotics derived in this study are being evaluated for their potential to modulate the human gut microbiota, and hopefully for further use as a novel medical technology for human gut microbiota rehabilitation.

## 2. Materials and Methods

### 2.1. Collection of Wild Edible Mushrooms

Wild mushrooms were collected from the local area of Ubon Ratchathani Province, Thailand. Mushrooms were kept in a collective paper bag and carried to the microbiology laboratory of the College of Medicine and Public Health, Ubon Ratchathani University, Warinchamrap, Ubon Ratchathani, Thailand. Subsequently, the fruiting body morphology of each mushroom was observed and recorded.

### 2.2. Genomic DNA Extraction

Fresh fruiting bodies of the mushrooms were selected for genomic DNA extraction. Approximately 100 mg of mushroom tissue was cut from a portion of the apical stalk of the fruiting bodies. The genomic DNA of mushrooms was extracted using the Prep Fungi/Yeast Genomic DNA Extraction Mini Kit according to the manufacturer’s instructions (Favorgen Biotech Corp., Ping Tung, Taiwan). The extracted DNA was stored at −20 °C until use.

### 2.3. Identification of Mushroom Species

The correct species of the collected mushrooms were identified by conventional polymerase chain reaction (PCR), DNA sequencing, and analysis, as previously described [25]. Genomic DNA extracted from each mushroom was used as the template. Table 1 shows the oligonucleotide primer pairs, LROR and LR6, and the PCR conditions used to amplify the target rRNA region of fungal genomic DNA, as previously reported [25]. All primers were purchased from Integrated DNA Technologies (Singapore). The AllTaq™ PCR Core Kit (Qiagen, Germantown, MD, USA) was used for PCR. The amplicons were verified through agarose gel electrophoresis and further purified using the HiYield Gel/PCR DNA Fragment Extraction Kit (RBC Bioscience, New Taipei City, Taiwan). DNA sequencing was performed at ATGC (Thailand Science Park, Pathumthani, Thailand) using the chain-termination DNA method. The amplicons were sequenced using the same primers used for the PCR. Nucleotide sequencing results were compared to those of fungal rDNA sequences obtained from the rRNA databases of the National Center for Biotechnology Information (NCBI) with highly similar sequence (megablast) parameter settings. A phylogenetic tree was constructed using MEGA version 11 [26]. The neighbor-joining method with 0.75 as the maximum sequence difference was used in this analysis.

### 2.4. Mushroom Preparation for Polysaccharide Extraction

The entire fruiting body of each mushroom was soaked in distilled water to remove contaminated soil and other debris. Cleaned mushrooms were cut into small pieces using scissors and were further incubated at 50 °C with forced air circulation for two days. Dried mushrooms were crushed into a fine powder using a blender (BL-T60 model, Thai Toshiba Electric Industries, Nonthaburi, Thailand). Mushroom powder (10 g) was added to 40 mL of absolute ethanol (VWR, BDH Chemical, Singapore) and incubated at ambient temperature with shaking at 150 rpm for 18 h. After incubation, the mixture was centrifuged at 9000× *g* for 20 min at 4 °C. The debris was collected and incubated in an oven at 50 °C for 24 h until completely dry.

### 2.5. Crude Polysaccharide Extraction

Dried ethanol-washed mushroom powder (10 g) was mixed with 90 mL of distilled water. The mixture was incubated at 90 °C in a water bath for 4 h. To collect the clear supernatant, the mixture was centrifuged three times at 9000× *g* for 20 min. The upper phase of the solution was collected and mixed with absolute ethanol (VWR, BDH Chemical) at a 1:4 ratio (upper phase supernatant:absolute ethanol), and the mixture was further incubated at 4 °C for 16–18 h. The mixture was centrifuged at 9000× *g* for 15 min. The pellet of extracted polysaccharide was washed with absolute ethanol and dried at 50 °C for 4 h. Thereafter, the polysaccharide was crushed into a fine powder using a mortar and pestle. The extraction yields were calculated using the following equation: $$Extraction\;yield=\left(\frac{g\;of\;crude\;polysaccharide}{g\;of\;dry\;weight\;of\;fruiting\;bodies}\right)\times 100$$

The extracted polysaccharide powder was stored in a bottle in a silica bag at 4 °C until use. 

### 2.6. Determination of Total Carbohydrate Content

The total carbohydrate content of the crude polysaccharide was determined as described previously [27]. Briefly, 50 μL of an appropriate dilution of crude polysaccharide was mixed with 150 μL of concentrated sulfuric acid (Fisher Scientific, Seoul, Republic of Korea) and was immediately mixed with 30 μL of 5% (*w*/*v*) phenol (Fisher Scientific). The mixture was maintained at 90 °C for 5 min. The reaction mixture was then cooled to ambient temperature. The absorbance of the reaction solution was measured at 490 nm using a microplate reader (Bio Chrom/EZ Read 2000, Biochrom Ltd., Waterbeach Cambridge, UK). A D-(+)-glucose solution (PanReac Applichem, Darmstadt, Germany) at different concentrations was used to calculate the standard concentration point.

### 2.7. Determination of Reducing Sugar

The reducing sugar present in the extracted crude polysaccharide was assayed using a 3,5-dinitrosalicyclic acid (DNS) (PanReac Applichem) assay, as previously described [28]. The absorbance of the reaction mixture was measured at 540 nm using a microplate reader. A D-glucose solution was used to calculate the standard concentration.

### 2.8. Determination of Protein Content

The protein content of the extracted crude polysaccharides was determined as previously described [9]. The absorbance of the reaction mixture was measured at 595 nm using a microplate reader. Bovine serum albumin (Sigma-Aldrich, Singapore) was used to prepare the standard protein solution.

### 2.9. Analysis of Monosaccharide Composition in Crude Polysaccharides

The crude polysaccharide powder was dissolved in sterile distilled water, and the pH was adjusted to 6.5. The solution was filtered using a 0.45 μM pore size filter (Whatman Puradisc^TM^, Buckinghamshire, UK). The monosaccharide composition of the crude polysaccharide solution was analyzed using HPLC at the Sugars and Derivatives Analytical Laboratory (SDAL), Kasetsart University, Bangkok, Thailand. Standard compounds glucose, mannose, galactose, and fucose were used for peak identification and concentration estimation. The reaction was performed in triplicate. 

### 2.10. Determination of Hydrolysis Tolerance in Simulated Gastric Buffer

The digestibility of the artificial gastric juice was evaluated as previously described [29]. Briefly, pH1 and pH5 of the hydrochloric acid buffer were used to simulate human gastric juice for the polysaccharide hydrolysis assay. The buffer was loaded into a polysaccharide solution (10 mg/mL) at an equal ratio and incubated at 37 °C for 2 h. Inulin was used as a comparative prebiotic. The reducing sugar released from the crude polysaccharide and the total sugar content were determined using the DNS assay and the phenol–sulfuric acid method, as previously described. The hydrolysis assay was performed after 0 and 2 h of incubation. The percentage hydrolysis was calculated using the following equation:$$\%\;Hydrolysis=\left(\frac{Reducing\;sugar\;at\;2\;h}{Total\;sugar\;content\;-\;Reducing\;sugar\;at\;0\;h}\right)\times 100$$

### 2.11. Antioxidant Activity by 1,1-Diphenyl-2-picrylhydrazyl Scavenging Ability

The scavenging ability was determined using 1,1-diphenyl-2-picrylhydrazyl (DPPH) radicals as described by Inyod et al. [30]. The DPPH solution was prepared to a final concentration of 0.2 mM DPPH (Sigma-Aldrich) in absolute ethyl alcohol (VWN). The crude polysaccharide samples were dissolved in distilled water at a concentration range of 0–5 mg/mL. The sample and DPPH solution were mixed in a 2:1 ratio and incubated for 30 min in the dark before the absorbance at 517 nm was measured using a spectrophotometer. The antioxidant activity was determined by measuring the 50% inhibitory concentration (IC_50_, mg/mL) of DPPH radicals. The percentage of inhibition was calculated using the following equation where ascorbic acid was used as the reference antioxidant:% Inhibition = (A_blank_ − A_sample_/A_blank_) × 100
where A_blank_ is the absorbance of the control reaction (containing all reagents except the test compound) and A_sample_ is the absorbance of the sample. 

Inhibitory activity values were calculated for various concentrations of the polysaccharide samples in triplicate.

### 2.12. Determination of Probiotic Bacterial Growth Promoted by Crude Polysaccharides

Three probiotic strains, *Limosilactobacillus fermentum* 47-7, *Lacticaseibacillus rhamnosus* NH9, and *Lacticaseibacillus rhamnosus* 23-2, were used to evaluate growth promotion by crudely extracted prebiotics. A single colony of each strain was inoculated into 10 mL of MRS broth and was incubated at 37 °C for 18 h. Bacterial cell pellets were harvested by centrifugation at 9000× *g* for 10 min at 4 °C. The bacterial cell pellets were washed twice with cold sterile distilled water. The bacterial cells were adjusted by comparing the bacterial solution with McFarland No. 1.0 (3.0 × 10^8^ CFU/mL). To collect bacterial cells, 200 μL of the adjusted bacterial cell solution was transferred into a microtube and centrifuged at 9000× *g* for 5 min. Bacterial cell pellets were collected and suspended with 200 μL MRS broth without glucose (MRSGlu−) supplemented with 20 mg/mL of crude polysaccharide, hereafter abbreviated as MRSGlu−/Pre. MRS with glucose (MRSGlu+) was used as a comparative medium. Fructo-oligosaccharide (FOS), galacto-oligosaccharide (GOS), and inulin (Inu) were purchased from Sigma and used as comparative prebiotics. The plates were incubated at 37 °C for 48 h. Bacterial growth was determined by measuring the turbidity in the wells at OD620 nm using a microplate reader 48 h after incubation. The experiment was performed in triplicate. The percentage bacterial growth for each MRSGlu/Pre ratio was calculated using the following equation:$$\%\;of\;bacterial\;growth=\left(\frac{OD620\;of\;MRSGlu-/Pre\;at\;48\;h}{OD620\;of\;MRS\;with\;glucose\;at\;48\;h}\right)\times 100$$

### 2.13. Preparation of Lyophilized Synbiotic

To prepare the lyophilized synbiotic, the *L. fermentum* strain 47-7 was used as a model probiotic and incorporated into the synbiotic component. *L. fermentum* 47-7 was grown in 10 mL MRS broth and was incubated at 37 °C for 18 h. Bacterial cell pellets were collected by centrifugation and washed twice with cooled sterile distilled water. Bacterial cell numbers were adjusted to obtain an approximate bacterial cell number of 1.2 × 10^9^ CFU/mL by comparing the bacterial solution with McFarland No. 4.0. Serial dilutions, followed by the plate count method, were used to confirm the viable bacterial cells. Bacterial cell pellets were harvested from 1 mL of the adjusted bacterial solution by centrifugation at 9000 rpm for 10 min. The bacterial cell pellet was suspended in 20 mL of a cryoprotectant medium, as shown in Table 2. The solution was kept on ice for 30 min and was further incubated at −20 °C for 18 h. The frozen solution was used to prepare a lyophilized powder using a freeze dryer (Alpha 1–4 LSCbasic, Martin Christ Company, Osterode am Harz, Germany) at the Faculty of Pharmacology, Ubon Ratchathani University, Ubon Ratchathani Province, Thailand.

### 2.14. Lyophilized Cells Observed Using Scanning Electron Microscopy

After lyophilization, each powder was aliquoted and placed in a tube covered with carbon tape. The lyophilized sample was coated with gold at a process current of 10 mA for 1 min (Sputter Coater, SPI module, West Chester, PA, USA). The morphology of the gold-coated samples was observed using scanning electron microscopy at 10.0 kv (JEOL JSM-6010 LV, Peabody, MA, USA) at the College of Medicine and Public Health, Ubon Ratchathani University, Ubon Ratchathani Province, Thailand.

### 2.15. Survival Ability of Probiotic during Synbiotic Storage 

The survival ability of the probiotic *L. fermentum* 47-7 was evaluated at 0, 30, 60, and 90 days of storage. Lyophilized powders from each sample were aliquoted from the storage tubes and weighed. The lyophilized powders were dissolved in 1 mL of sterile distilled water and used to prepare 10-fold dilutions. Next, 100 μL of an appropriate dilution was spread onto MRS agar plates. After 48 h of incubation at 37 °C, colonies were counted and calculated as colony-forming units (CFU per mL). The experiment was performed in triplicate. 

## 3. Results

### 3.1. Wild Mushroom Collection and Fruiting Body Morphology Recording

In the present study, two mushroom samples, UBU_LS1 and UBU_LP2, were collected separately from deadwood logs. UBU_LS1 was collected from a forest in Phibun Mangsahan district, Ubon Ratchathani Province, Thailand, at the following geographic coordinates: 15°13′47.4″ N 105°13′22.3″ E. UBU_LP2 was collected from Nong Ejem swamp in Ubon Ratchathani University, Warinchamrab district, Ubon Ratchathani Province, Thailand, at the following geographic coordinates: 15°07′43.0″ N 104°54′46.2″ E. Observation of fruiting body morphology revealed that UBU_LS1 had white and umbilicated sporocarps (Figure 1a), while UBU_LP2 had funnel-like brown-to-fuscous lamellae (Figure 1b).

### 3.2. Molecular Identification of Mushroom Species

Species identification was carried out using a molecular-based technique in which a large subunit of the 28S ribosomal RNA gene (LSU-28S rRNA) was amplified, sequenced, and analyzed. As shown in Figure 2, an amplified product of approximately 1000 base pairs (bp) was obtained from both UBU_LS1 and UBU_LP2, and after nucleotide sequencing, it was found that these products contained 1050 and 1016 bp, respectively. Sequence comparison of the LSU-28S rRNA gene of UBU_LS1 with those held in the NCBI database demonstrated that the LSU-28S rRNA gene of UBU_LS1 had an identity similarity (100%, 1050/1050 identities) with the LSU-28S rRNA gene of *L. squarrosulus* FRIM4180 (GenBank accession No. KP283517.1). For UBU_LP2 LSU-28S rRNA gene analysis, we observed an identity similarity (100%, 1016/1016 identities) with *Lentinus polychrous* KM141387 (GenBank accession No. KP283514.1). Phylogenetic tree analysis of the LSU-28S rRNA gene revealed that UBU_LP2 and UBU_LS1 were closely related to *L. polychrous* and *L. squarrosulus,* respectively (Figure 3). Therefore, based on morphological appearance and rRNA gene sequencing and analysis, the two wild mushroom isolates, UBU_LS1 and UBU_LP2, were identified as *Lentinus squarrosulus* and *Lentinus polychrous*, respectively. 

### 3.3. Deposition of Ribosomal RNA Gene Sequence in the NCBI Database

The 28S large subunit ribosomal RNA gene sequences of UBU_LS1 and UBU_LP2 were deposited in the National Center for Biotechnology Information (NCBI) database under accession numbers OQ915355 and OQ915356, respectively.

### 3.4. Yields of Crude Polysaccharide Extraction

After extraction, the crude polysaccharides derived from *L. squarrosulus* UBU_LS1 (hereafter referred to as CPS_UBU_LS1) and *L. polychrous* UBU_LP2 (hereafter referred to as CPS_UBU_LP2) were dark brown in color (Figure 4). It was found that the extraction yield of CPS_UBU_LS1 and CPS_UBU_LP2 was 6.77 ± 0.02% and 7.14 ± 0.01% (Table 3), respectively.

### 3.5. Compound Component of Extracted Crude Polysaccharide 

Table 3 lists the components of CPS_UBU_LS1 and CPS_UBU_LP2. The major components of both CPS_UBU_LS1 and CPS_UBU_LP2 were carbohydrates, and there were low amounts of protein and reducing sugars. CPS_UBU_LS1 and CPS_UBU_LP2 had high polysaccharide contents, which were approximately calculated as 93.53% and 92.61% of total carbohydrates.

### 3.6. Monosaccharide Composition of CPS_UBU_LS1 and CPS_UBU_LP2

The monosaccharide composition of mushroom extracts quantified by HPLC is shown in Table 4. It was found that polysaccharides of both CPS_UBU_LS1 and CPS_UBU_LP2 were composed of monosaccharides in varying concentrations. Fucose was found to be the most abundant monosaccharide in both CPS_UBU_LS1 and CPS_UBU_LP2 crude extracts, followed by mannose, glucose, and galactose at varying concentrations. These results indicated that the polysaccharides extracted from both UBU_LS1 and UBU_LP2 using hot water were heterologous polysaccharides. 

### 3.7. Resistance in Simulated Human Gastric Acidic pH Solution

Figure 5 shows that all crude polysaccharides, including the prebiotic inulin, were highly resistant to hydrolysis at both pH1 and pH5 after 2 h of incubation. The results demonstrated that the crude polysaccharides of both CPS_UBU_LS1 and CPS_UBU_LP2 had a high hydrolysis tolerance of more than 90%. The crude polysaccharide of CPS_UBU_LP2 showed the highest hydrolysis tolerance at 95% at both pH1 and pH5, which was higher than that of CPS_UBU_LS1 and inulin.

### 3.8. Potential of Crude Polysaccharides for Stimulating Probiotic Growth

Figure 6 shows the potential of all crude polysaccharides, including comparative prebiotics (inulin, fructo-oligosaccharide (FOS), and galacto-oligosaccharide (GOS)), for stimulating the probiotic strains, 47-7, 23-2, and NH9. The percentage of growth of each strain was compared to that derived from MRS medium supplemented with glucose, which was taken as 100%. It was found that the crude polysaccharide CPS_UBU_LP2 had a high potential for stimulating growth in all probiotic strains, especially in the *L. fermentum* 47-7 strain, for which stimulation with CPS_UBU_LP2 was not significantly different from stimulation with GOS. Moreover, the crude polysaccharides CPS_UBU_LP2 and CPS_UBU_LS1 had the potential to stimulate all probiotic strains to significantly higher levels than that observed with stimulation by inulin and FOS.

### 3.9. Antioxidant Activities of Crude Polysaccharides

The antioxidant activities of the crude polysaccharides CPS_UBU_LS1 and CPS_UBU_LP2 were evaluated using DPPH radical scavenging assays. The antioxidant effectiveness was expressed as the IC_50_. As shown in Figure 7, the result indicated that CPS_UBU_LP2 and CPS_UBU_LS1 were potent DPPH scavengers with IC_50_ values of 0.38 and 0.58 mg/mL, respectively. The control (ascorbic acid) showed the highest antioxidant activity, with an IC_50_ value of 0.01 mg/mL. 

### 3.10. Potential Crude Polysaccharides as Cryoprotectants in the Freeze-Drying Process and for Storage

Crude polysaccharides CPS_UBU_LS1 and CPS_UBU_LP2 were used to formulate lyophilized synbiotic products, and *L. fermentum* 47-7 was selected as a model probiotic. After freeze drying, bacterial viability at 0, 30, 60, and 90 days was determined using the 10-fold dilution and plate count technique. The cell viability was expressed as colony-forming units (CFU) per ml. At 0 days (immediately after freeze drying), it was shown that all lyophilized cells had a high survival percentage of more than 90%, even for the lyophilized cells without a cryoprotectant (Figure 8). The greatest survival (99.90%) on day 0 was derived for freeze-dried 47-7 formulated with skim milk (SM) as the cryoprotectant (47-7+SM). After 90 days of storage at 4 °C, all freeze-dried cells exhibited a slow reduction in viability up to 90 days of storage. The freeze-dried *L. fermentum* 47-7 with a cryoprotectant derived from CPS_UBU_LP2 and CPS_UBU_LS1 exhibited viability of 77.64% (1.55 × 10^10^ CFU/mL) and 78.40% (1.57 × 10^10^ CFU/mL), respectively. Moreover, we found that the viability of freeze-dried 47-7 formulated with crude polysaccharides CPS_UBU_LP2 and CPS_UBU_LS1 was slightly enhanced upon the addition of SM to the formula (data are shown in Appendix A).

### 3.11. Lyophilized Cell Morphology Observed Using SEM

Scanning electron micrographs of *L. fermentum* 47-7 in different cryoprotective media are shown in Figure 9. The cryoprotective media of skim milk (Figure 9b) and crude polysaccharides, CPS_UBU-LS1 (Figure 9d) and CPS_UBU-LP2 (Figure 9e), enveloped the probiotic *L. fermentum* 47-7 inside the lyophilized powder. In addition, the reference prebiotic GOS had a smooth surface that coated the bacterial cells (Figure 9c). The lyophilized *L. fermentum* 47-7 cells without any cryoprotectant showed some damage presenting as pores in bacterial cells (Figure 9a). These SEM images confirmed that all wall materials facilitated the viability of probiotic cells during the lyophilization process at high temperatures and pressures.

## 4. Discussion

In this study, wild edible mushrooms were collected from a local area in Ubon Ratchathani Province, Thailand, and their properties and applications as cryoprotectants in the freeze-drying process for synbiotic product development were determined. However, there is serious evidence that eating wild mushrooms without correctly distinguishing between poisonous and edible mushrooms may cause illness or death [31]. Thus, not only the morphological identification of fresh fruiting bodies, but also a reliable method for the precise identification of mushrooms must be developed to ensure that mushroom samples are edible and safe. The molecular method of PCR followed by nucleotide sequence analysis, as previously described by Raja et al. (2017), was used in our study [25]. Our results demonstrated that the LROR/LR6 primers used to analyze the nucleotide regions of the 28S rRNA large subunit accurately identified the wild mushroom sample isolates UBU_LS1 and UBU_LP2 as *L. squarrosulus* and *L. polychrous*, respectively. 

Currently, a wide range of polysaccharide extraction methods are available for different mushroom species. These methods include conventional hot water extraction (HWA), subcritical water extraction (SWE), enzyme-assisted extraction (EAE), ultrasonic-assisted extraction (UAE), microwave-assisted extraction (MAE), ultrasonic–microwave synergistic extraction (UMSE), pulsed electric field-assisted extraction (PEFAE), and aqueous two-phase extraction (ATPE) [32]. Among these methods, HWA is widely used because of its simplicity and because it does not require specialized equipment. Herein, polysaccharides derived from the dried fruiting bodies of *L. squarrosulus* UBU_LS1 and *L. polychrous* UBU_LP2 were extracted using HWA, followed by absolute ethanol precipitation. The extraction yield of crude polysaccharides from *L. polychrous* UBU_LP2 was slightly higher than that from *L. squarrosulus* UBU_LS1. This was attributed to species-dependent factors. Our results are in agreement with a previous report by Nowak et al., who demonstrated that using the same protocol for different species resulted in variable yields [33]. Thus, a species-dependent factor affects the polysaccharide extraction yield. The extraction yield of both *L. squarrosulus* UBU_LS1 and *L. polychrous* UBU_LP2 exceeded the previous report that used the same mushroom species and different species such as *Grifola frondose* (3.35 ± 0.16%) [29] and *Inonotus obliquus* (3.81 ± 0.34%) [34]. However, it has been demonstrated that the extraction yield derived using the HWE method may be increased if the protocol is optimized. As previously reported for polysaccharide extraction from wild *G. frondose*, a higher extraction yield was obtained when the extraction temperature and time and the ratio of liquid to raw material were optimized [35]. Moreover, repeated extractions (up to six) enhanced the polysaccharide yield without altering polysaccharide properties [32].

In addition to the quantity of extracted polysaccharides, the composition of crude polysaccharides also influences their properties [36]. Both CPS_UBU_LS1 and CPS_UBU_LP2 contained a high carbohydrate content, which was mostly polysaccharides, and very low amounts of reducing sugars and proteins. Thus, the crude extracts derived from both mushrooms could be considered as crude polysaccharides. High-performance liquid chromatography (HPLC) analysis of the monosaccharide components revealed that CPS_UBU_LS1 and CPS_UBU_LP2, composed of glucose, galactose, fucose, and mannose with varying concentrations of these fucoses, were greater, indicating a typical heteropolysaccharide. It has been reported that crude polysaccharides from *L. squarrosulus* extracted by HWA are composed of galactose, glucose, and mannose as the major monosaccharides [37]. Using a similar HWA extraction method, the crude polysaccharide content of *Agaricus bisporus* and *Agaricus brasiliensis* was mainly constituted of glucose, mannose, galactose, and methyl-galactose, with low amounts of fucose and ribose. Among these monosaccharides, fucose, in a formulation with fucogalactan, is presumed to play an important role in immunomodulatory activities [38]. Moreover, Thetsrimuang et al. (2011) found that mannose was a major monosaccharide in crude polysaccharides extracted from *Lentinus* sp. [39]. Mannose is an essential monosaccharide for biological activities, such as antioxidation, in various polysaccharides, including those of *Lentinus* species [9,40]. Xylose has been reported as a minor monosaccharide that may be an essential component of biological properties in *L. squarrosulus* crude polysaccharide extracts [38]. Thus, CPS_UBU_LS1 and CPS_UBU_LP2 were reasonable for further elucidation of their properties, including the influence of polysaccharide composition and structure on their biological activities. 

Prebiotics are attractive candidates comprising crude and pure polysaccharides extracted from edible mushrooms. Some edible cultivatable and wild mushrooms have been examined for their prebiotic potential and other properties, such as antioxidant activity, cytotoxicity against cancer cells [9], and pathogenic growth inhibition [40]. Two types of well-characterized prebiotic fructans (inulin and fructo-oligosaccharide (FOS)) and galacto-oligosaccharides (GOSs) have been used for health promotion and for treating some gastrointestinal disorders [41]. Another major goal of using prebiotics in humans is to modulate beneficial microorganisms, mainly bacteria, in the large intestine [42]. Thus, to reach the large intestine, prebiotics must resist the acidic conditions of the stomach and the enzymatic system in the intestine. Our study demonstrated that CPS_UBU_LS1 and CPS_UBU_LP2 had resisted hydrolysis by simulated human gastric acidic solution at both pH 1 and pH 5, as polysaccharide activity was retained at more than 90%. The retained polysaccharide activities of these two crude prebiotics were comparable to those of inulin, which was used as the reference prebiotic. These results indicated that CPS_UBU_LS1 and CPS_UBU_LP2 are suitable for application as prebiotics in humans.

In addition to hydrolytic resistance under acidic conditions, stimulation of beneficial bacteria, particularly probiotic strains, is a major property of candidate prebiotics [2]. CPS_UBU _LS1 and CPS_UBU_LP2 stimulated three probiotic strains, *L. fermentum* 47-7, *L. rhamnosus* NH9, and *L. rhamnosus* 23-2. The percentage of growth stimulation of these probiotic strains with CPS_UBU_LS1 and CPS_UBU_LP2 was significantly higher than that of inulin and FOS and comparable to that of GOS. Our results are in concordance with previous findings by Nowak et al., who found that crude polysaccharides extracted from wild edible mushrooms could act as prebiotics that stimulate the growth of probiotic bacterial strains [33]. Other mushrooms, such as *Pleurotus ostreatus*, *Pleurotus eryngii*, and *Trametes versicolor*, reportedly contain polysaccharides that stimulate probiotic lactobacilli and *Bifidobacterium* growth [43,44]. Moreover, the crude polysaccharide and its purified fraction from the medicinal mushroom *Ganoderma lucidum* have potential as prebiotics to stimulate *Bifidobacterium* species in in vitro and human fecal fermentation assays [45]. Thus, based on this evidence, crude mushrooms can be used as a source of prebiotics, and further studies are required to evaluate their potential in vivo, especially in healthy humans and/or those with particular diseases.

In the DPPH radical scavenging assay, a low IC_50_ value indicates a potent DPPH scavenger. The crude polysaccharides CPS_UBU_LS1 and CPS_UBU_LP2 could be considered as potent antioxidative compounds as they showed low IC_50_ values of 0.5 and 0.38, respectively. These values were higher than those of ascorbic acid, a reference antioxidant compound. The potent DPPH scavenging activity of the polysaccharides extracted from *L. squarrosulus* was enhanced by increasing the concentration of polysaccharides in the reaction. This indicates the dose-dependent properties of polysaccharides as DPPH scavengers [38]. Moreover, it has been reported that the antioxidant activity of polysaccharides could be enhanced by the chemical modification of polysaccharide structure, for example by sulfation, resulting in a high solubility of crude polysaccharides and leading to greater antioxidant activity [46]. As detailed in a previous report, the polysaccharide properties, including solubility, net charge, and side chain, play an important role in their antioxidant activity [47]. Thus, further analysis of CPS_UBU_LS1 and CPS_UBU_LP2 properties, as mentioned above, is required to understand their function. 

Many commercial functional food products, particularly synbiotics, have attracted considerable consumer attention. The final synbiotic product is mostly non-encapsulated, encapsulated, or pilled. Thus, the production chain and storage are crucial processes. Freeze drying or lyophilization is a widely used method for preserving viable bacteria, including probiotics, in a dried form [24]. This process requires two main components: the target probiotic strain and wall material or one or more effective cryoprotectants. Some cryoprotectants, such as monosaccharides (glucose, galactose, and mannose), disaccharides (trehalose, sucrose, turanose, and lactose), standard prebiotics (FOS, GOS, and lactulose), and other substances including skim milk, have been used to enhance the survival of specific probiotics during freeze drying [48,49,50]. According to the definition of synbiotics, the compounds acting as cryoprotectants used in synbiotic formulations must be “food-grade” [51]. Freeze drying does not affect the antioxidant activity of crude polysaccharides extracted from the mushroom *I. obliquus* [52]. Moreover, it was reported that freeze drying did not alter the quality, including the polysaccharide properties, of products derived from the mushroom *Tremella fuciformis* [53]. Thus, polysaccharides provide an attractive option for use as cryoprotectants. Our result showed that CPS_UBU_LS1 and CPS_UBU_LP2 preserved the survival ability of the selected probiotic *L. fermentum* 47-7 for up to 90 days of storage at 4 °C at the viability of 1.55 × 10^10^ CFU/mL and 1.57 × 10^10^ CFU/mL, respectively. The number of viable cells was comparable to that obtained from lyophilized cells treated with SM and GOS. Additionally, the number of viable cells in our study showed adequate probiotic benefits [54]. It has been reported that monosaccharides, such as glucose and mannose, found in crude polysaccharide played an importance role in the viability during freeze drying and storage. The lyophilized cells were enveloped inside the cryoprotectant, which may be the protective mechanism by which the crude polysaccharide preserves probiotics, as damaged cells were not observed under SEM in the samples, except for lyophilized 47-7 cells alone. 

## 5. Conclusions

In this study, two wild edible mushrooms collected from the local area of Ubon Ratchathani Province, Thailand, were successfully identified at the species level. We demonstrated that wild edible mushrooms of *L. squarrosulus* and *L. polychrous* can be a natural source of prebiotics and applied as cryoprotectants in the freeze-drying process for the development of a synbiotic product. Moreover, these crude polysaccharides exhibited antioxidative potential. Although our result demonstrated the potential prebiotic of crude polysaccharides extracted from wild edible mushrooms, the prebiotic properties need to be evaluated ex vivo and in vivo in the next research steps. Further evaluation of these prebiotics, including the elucidation of the polysaccharide structure–function relationship and the determination of the potential rehabilitation of beneficial gut microbes in diseased individuals, is currently being conducted by our research group.

## Figures and Tables

**Figure 1 foods-13-00287-f001:**
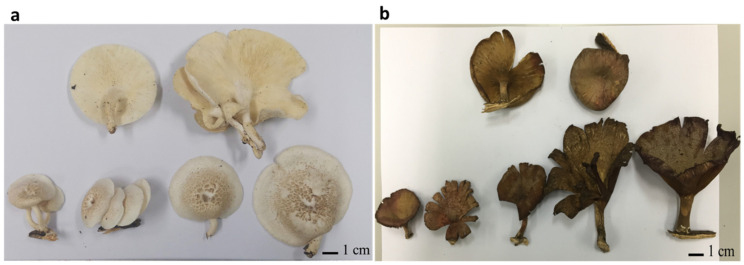
Morphology of fresh fruiting bodies of *Lentinus squarrosulus* UBU_LS1 (**a**) and *Lentinus polychrous* UBU_LP2 (**b**). Scale bar represents 1 centimeter (cm).

**Figure 2 foods-13-00287-f002:**
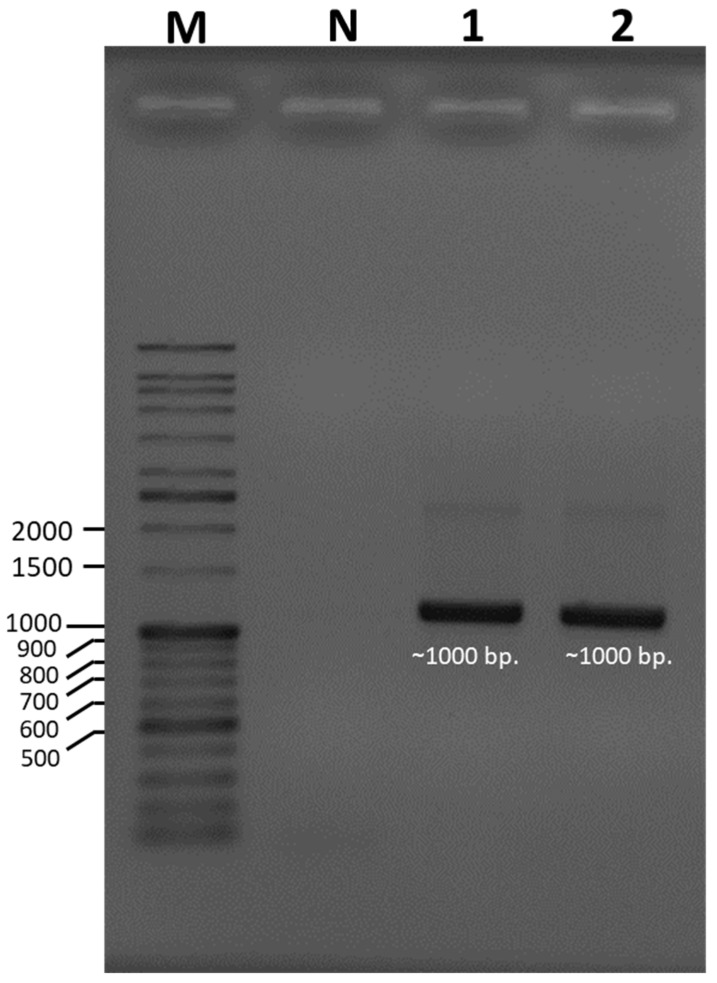
Ethidium bromide-stained agarose gel showing the amplified product of a large subunit of the 28S ribosomal RNA gene (LSU-28S rRNA) of *Lentinus squarrosulus* UBU_LS1 and *Lentinus polychrous* UBU_LP2. Lane M contained the VC DNA ladder. Lanes 1 and 2 contained the amplified products of the LSU-28S rRNA genes of UBU_LS1 and UBU_LP2, respectively. Lane N contained the PCR product without any DNA template. The nucleotide size (base pairs, bp) of the DNA ladder is shown beside the gel.

**Figure 3 foods-13-00287-f003:**
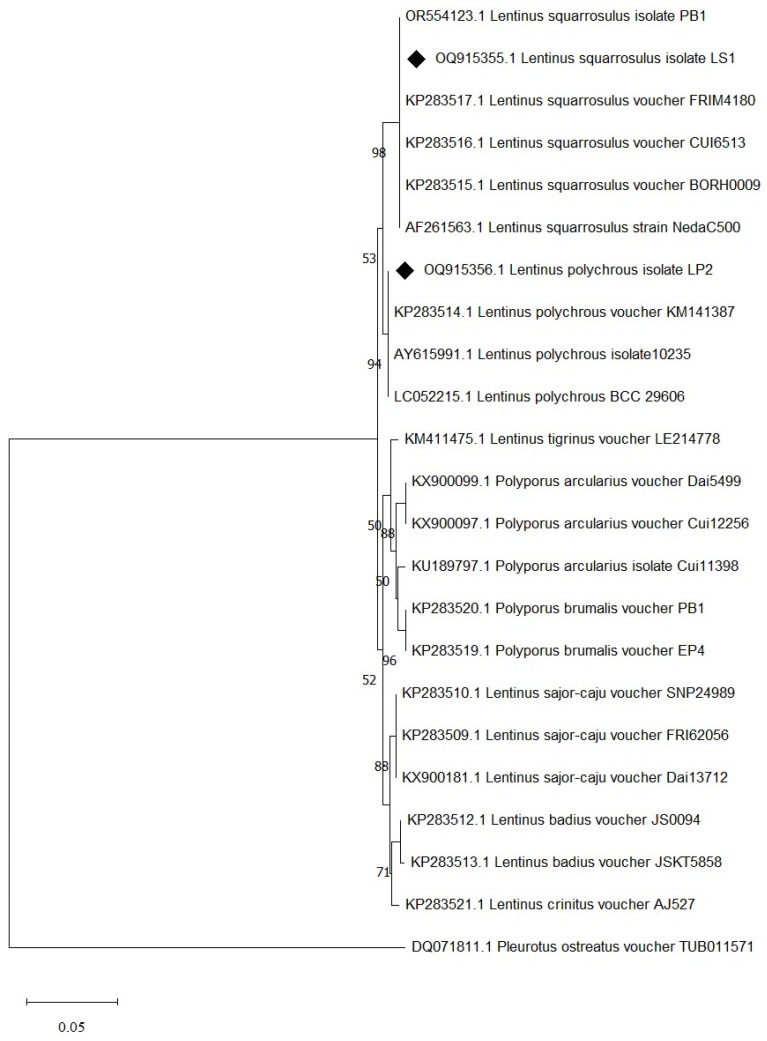
Evolutionary relationships of taxa inferred from the neighbor-joining method using LSU-28S rRNA gene data. Bootstrap support value is greater than or equal to 50% is shown in each node. Evolutionary analyses were conducted using MEGA11. *Pleurotus ostreatus* was used as the outgroup. The LSU-28S rRNA of *L. squarrosulus* UBU_LS1 and *L. polychrous* UBU_LP2 are indicated by the symbol ♦ before their accession number.

**Figure 4 foods-13-00287-f004:**
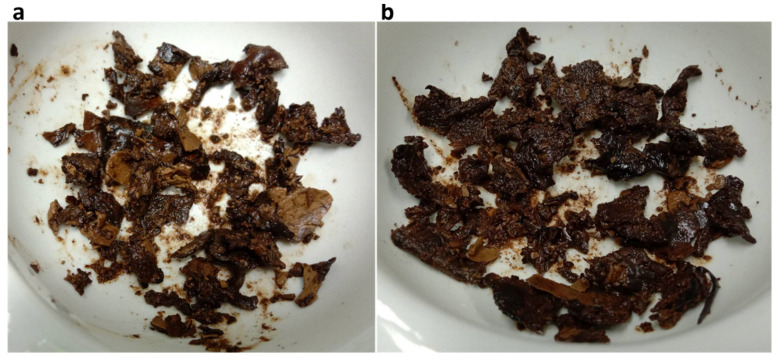
Appearance of crude polysaccharide CPS_UBU_LS1 (**a**) and CPS_UBU_LP2 (**b**).

**Figure 5 foods-13-00287-f005:**
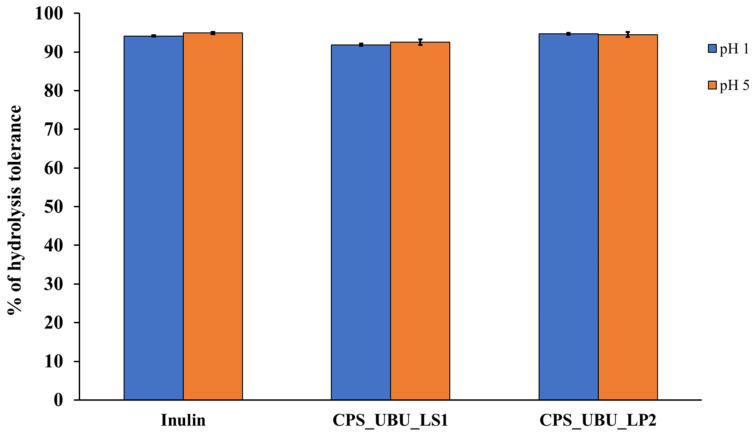
The percentage of hydrolysis tolerance of crude polysaccharides CPS_UBU_LS1 and CPS_UBU_LP2. Inulin was used as reference prebiotic. (Raw data is shown in Supplement data for Figure 5 (Appendix A)).

**Figure 6 foods-13-00287-f006:**
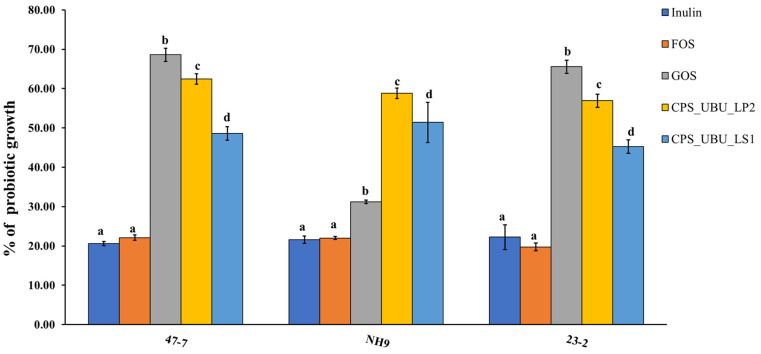
Growth stimulation of probiotic bacteria by crude polysaccharides CPS_UBU_LS1 and CPS_UBU_LP2. Inulin, FOS, and GOS were used as reference prebiotics. Different letters indicate a significant difference (*p* < 0.05).

**Figure 7 foods-13-00287-f007:**
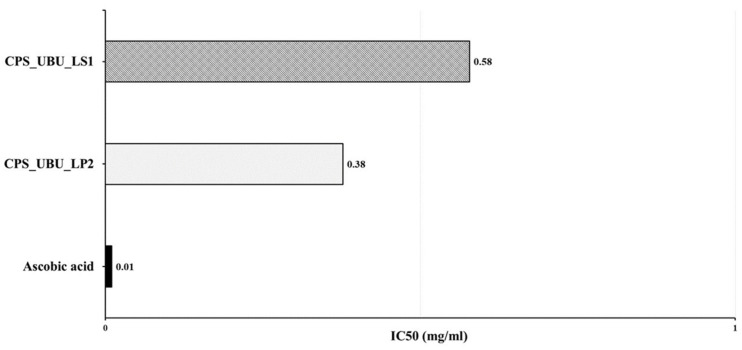
IC50 values of crude polysaccharides CPS_UBU_LS1 and CPS_UBU_LP2 measured in DPPH assay. Ascorbic acid was used as reference antioxidant.

**Figure 8 foods-13-00287-f008:**
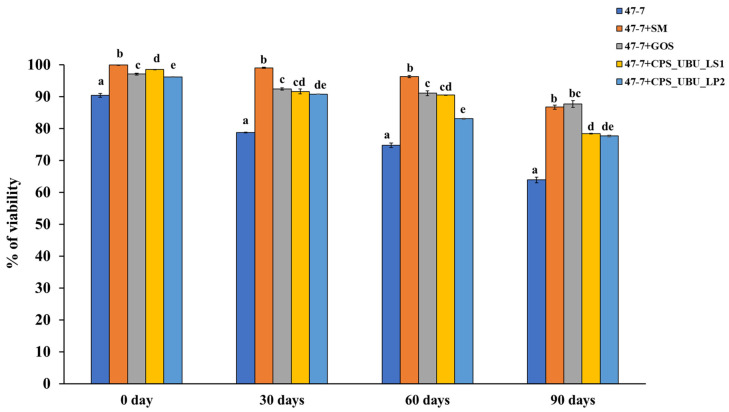
Viability of probiotic *L. fermentum* 47-7 formulated with and without prebiotic after freeze drying and storage for 90 days. The small letters indicate significant differences (*p* < 0.05) from the control (without prebiotic).

**Figure 9 foods-13-00287-f009:**
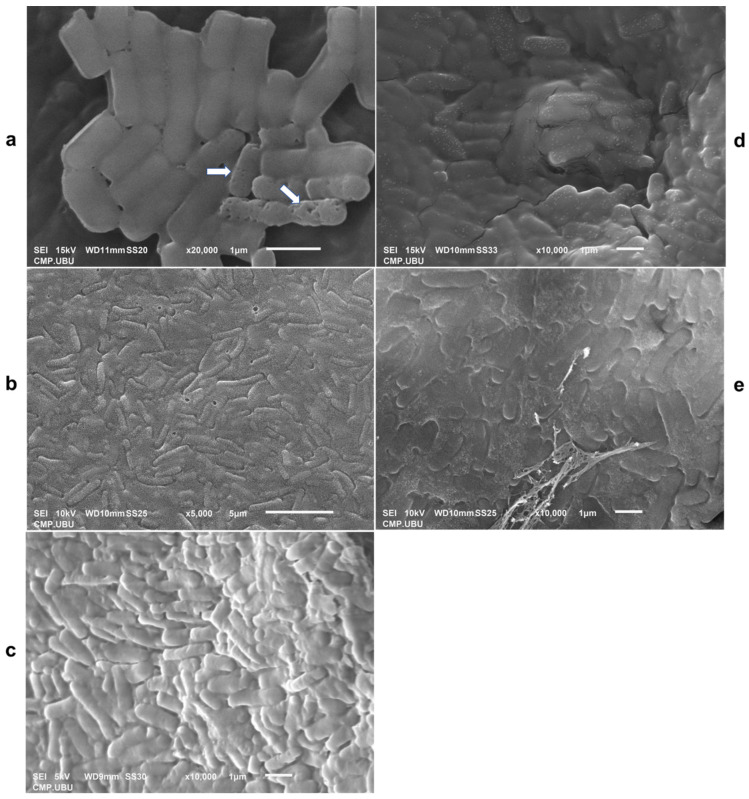
Scanning electron micrographs of lyophilized cells of *L. fermentum* 47-7 without cryoprotective media (**a**) and with skim milk (SM) (**b**), galacto-oligosaccharide (GOS) (**c**), CPS_UBU_LS1 (**d**), and CPS_UBU_LP2 (**e**). The damaged bacterial cells are indicated by arrows.

**Table 1 foods-13-00287-t001:** Oligonucleotide primers and PCR conditions used in this study.

Primer	Amplified DNA Size (bp.)	Sequence (5′→3′)	PCR Condition	Reference
Large subunit (LSU, 28S) of the rRNA				
LROR	~1000 bp	ACCCGCTGAACTTAAGC	Step 1: 1 cycle of 95 °C, 5 min Step 2: 35 cycles of 94 °C, 30 s; 52 °C, 30 s; 72 °C, 1 min Step 3: 1 cycle of 72 °C, 8 min	[25]
LR6		CGCCAGTTCTGCTTACC		

**Table 2 foods-13-00287-t002:** Ingredients of cryoprotective media used for preparation of lyophilized synbiotic in freeze drying.

Cryoprotective Media	*L. fermentum* 47-7 (CFU/mL)	Ingredients (mg)			
		Skim Milk	GOS	CPS_UBU_LS1	CPS_UBU_LP2
47-7	2 × 10^10^	-	-	-	-
47-7+SM	2 × 10^10^	500	-	-	-
47-7+GOS	2 × 10^10^	-	500	-	-
47-7+CPS_UBU_LS1	2 × 10^10^	-	-	500	-
47-7+CPS_UBU_LP2	2 × 10^10^	-	-	-	500

**Table 3 foods-13-00287-t003:** Yield of crude polysaccharides and the amount of total carbohydrates, reducing sugars, polysaccharides, and protein present in crude polysaccharides CPS_UBU_LS1 and CPS_UBU_LP2 (*n* = 3).

Crude Polysaccharide	Yield ^a^ (%)	Total Carbohydrate (mg/g)	Reducing Sugar (mg/g)	Polysaccharide (mg/g)	Total Protein (mg/g)
CPS_UBU_LS1	6.77 ± 0.02	313.99 ± 0.83	20.28 ± 1.92	293.68 ± 4.59	28.23 ± 0.38
CPS_UBU_LP2	7.14 ± 0.01	364.65 ± 1.68	26.94 ± 0.96	337.71 ± 7.53	31.46 ± 0.36

^a^ Yield per hundred grams of dry sample weight. The value is expressed as mean ± S.D. of triplicate experiments.

**Table 4 foods-13-00287-t004:** Monosaccharide composition in crude polysaccharides CPS_UBU_LS1 and CPS_UBU_LP2.

Crude Polysaccharide	Glucose (mM)	Galactose (mM)	Fucose (mM)	Mannose (mM)
CPS_UBU_LS1	0.88	0.22	2.13	1.10
CPS_UBU_LP2	1.13	0.11	2.04	1.02

## Data Availability

Data is contained within the article or Appendix A.

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
