# Peer review of "Evaluation of Prebiotic Potential of Crude Polysaccharides Extracted from Wild Lentinus polychrous and Lentinus squarrosulus and Their Application for a Formulation of a Novel Lyophilized Synbiotic"

_foods, 2024, doi:10.3390/foods13020287_

Round 1
Reviewer 1 Report
Comments and Suggestions for Authors
The authors collected, sequenced and identified mushroom species.
Hot water concentration (HWA) extractions were carried out. Hydrolysis and antioxidant activities were calculated. The cryoprotective properties were also observed by microscopy.
Some questions may be more enlightening:
(a) The authors found glucose, galactose, fucose and mannose in the mM range. Is there any other saccharide in lower concentration that could also contribute to the properties of mushrooms?
(b) Did the authors confirm the monosaccharides by some other technique such NMR spectroscopy?
(c) Who contributes to the properties (antioxidant activities and cryoprotective properties) of mushrooms? Just the polysaccharides? As fucose is the most concentrated species, should it be attributed the properties of mushrooms, or should a combination of the entire composition be responsible for this?
Reviewer 2 Report
Comments and Suggestions for Authors
1. The conclusion of the abstract should provide specific details on the role and significance of this study.
2. Suggest increasing the prebiotic activity of polysaccharides in the introduction of the preface.
3. Why did shiitake mushroom polysaccharides not undergo decolorization and deproteinization treatment.
4. two mushroom samples, UBU_LS1 and UBU_LP2: Are the two materials selected by the author representative.
5. total carbohydrate, reducing sugar, polysaccharide, total protein. The above indicators are recommended to be compared horizontally and vertically with published articles that belong to the same category.
6. Fig. 5, Suggest the author to provide raw data for reviewers to review. Fig. 6 and Fig. 8, Suggest adding significant difference markers.
7. Suggest increasing the discussion and analysis of the structure-activity relationship between structure and prebiotic activity.
8. The format of references needs to be standardized, and it is recommended that authors cite it: 10.1007/s11694-022-01288-3.
Reviewer 3 Report
Comments and Suggestions for Authors
Dear authors,
Crude polysaccharides extracted from wild Lentinus polychrous and Lentinus squarrosulus refer to the unrefined, natural extracts of polysaccharides obtained from these specific mushroom species. There is scientific interest in the potential health benefits of mushroom polysaccharides, the field is dynamic, and research is ongoing.
Research on crude polysaccharides from mushrooms often involves studying their biological activities and potential health benefits. The specific effects and applications of crude polysaccharides from Lentinus polychrous and Lentinus squarrosulus depends on the outcomes of specific studies and the compounds present in these extracts.
Therefore the present study is going beyond state-of-the-art and is interesting.
Comments on the Quality of English LanguageMinor spelling
